# Transmissible Gastroenteritis Virus Binding to Red Blood Cells Disrupts Iron Homeostasis and Promotes Viral Infection

**DOI:** 10.3390/vetsci13010042

**Published:** 2026-01-03

**Authors:** Lu Xia, Ziqi Wang, Yeqing He, Jingwen Wang, Junyuan Ren, Erhao Zhang, Zhonghu Liu, Yilei Li, Zi Li, Zhanyong Wei

**Affiliations:** 1College of Veterinary Medicine, Henan Agricultural University, Zhengzhou 450046, China; 2Ministry of Education Key Laboratory for Animal Pathogens and Biosafety, Zhengzhou 450046, China; 3College of Veterinary Medicine, Jilin University, Changchun 130062, China

**Keywords:** transmissible gastroenteritis virus, red blood cells, bind, oxygen-carrying, erythrophagocytosis, iron

## Abstract

Transmissible gastroenteritis virus (TGEV) is an enteric coronavirus, and numerous studies have focused on the interaction between TGEV and the intestinal tract. In our experiment, breathlessness was observed in TGEV-infected piglets, suggesting impaired oxygen uptake by RBCs. The effects of TGEV infection on red blood cells (RBCs) and its pathophysiological implications were explored. TGEV was found to bind to, but not replicate in, RBCs. TGEV binding to RBCs causes membrane structure damage, impairs oxygen-carrying capacity, promotes macrophage-mediated phagocytosis of damaged RBCs, and reduces serum iron levels, thereby facilitating viral infection. These findings highlight the importance of diagnostic technologies targeting RBCs and suggest new avenues for drug development and vaccine design for TGEV prevention. Further exploration of the mechanisms by which RBCs contribute to viral infection is warranted.

## 1. Introduction

Transmissible gastroenteritis virus (TGEV) is a major enteric coronavirus responsible for vomiting, diarrhea, and dehydration. The virus was first discovered in the United States in 1946 and subsequently spread to Europe, Asia, Africa, and South America. All age groups of pigs are susceptible to TGEV infection, whereas morbidity and mortality in piglets younger than 2 weeks old can approach nearly 100% [1]. Surviving pigs usually show a relatively low growth rate or even become “stiff pigs”, resulting in significant economic losses to the swine industry. Although epidemiological surveys reported a decline in the prevalence of TGEV in recent years, the emergence of variant strains and coinfections with other enteric viruses has exacerbated disease severity [2,3,4]. Numerous studies have focused on the interaction between TGEV and the intestinal tract. The blood system plays critical roles in disease pathophysiology and mortality; however, its involvement in enterovirus infections remains unclear. In our clinical observations, piglets infected with TGEV exhibited not only diarrhea but also dyspnea, indicating impaired oxygen delivery to the lungs. These findings led us to hypothesize that TGEV disrupts the structure and function of red blood cells (RBCs).

RBCs are the most abundant cells in the blood and play a critical role in oxygen transport. Due to their lack of nucleus or organelles, mature mammal RBCs have long been considered incapable of supporting viral replication. However, accumulating evidence indicates that RBCs can interact with various viruses and contribute to viral pathogenesis. RBCs have been shown to bind human immunodeficiency virus (HIV-1), influenza A virus (IAV), adenovirus, Zika virus, and severe acute respiratory syndrome coronavirus 2 (SARS-CoV-2), suggesting that RBCs may serve as reservoirs [5,6,7]. Data showed that viral infection damages RBCs, leading to their recognition and clearance by macrophages. During erythrophagocytosis, iron is released and recycled for erythropoiesis. However, the underlying mechanisms driving these physiological and pathological changes during viral infection remain largely unexplored.

In the present study, we investigated the effects of TGEV infection on RBCs and its pathophysiological implications. Our results demonstrated that TGEV interacts with RBCs in a manner similar to other pathogens. Infection with TGEV resulted in significant alterations to hematological parameters in piglets. Further analyses revealed that TGEV binds to RBCs, causing membrane structure damage, impairing oxygen-carrying capacity, promoting macrophage-mediated phagocytosis of damaged RBCs, and reducing serum iron levels, thereby facilitating viral infection. These findings enhance our understanding of the progression of TGEV infection and provide novel insights into strategies for its prevention and vaccine design.

## 2. Materials and Methods

### 2.1. Cells and Viruses

Swine testicle (ST) cells and immortalized porcine alveolar macrophages (3D4/21) were purchased from the China Institute of Veterinary Drug Control. Cells were supplied in Dulbecco’s modified Eagle’s medium (DMEM; Gibco, Waltham, MA, USA) supplemented with 10% fetal bovine serum (FBS, Gibco) at 37 °C in a 5% CO_2_ humidified atmosphere. RBCs were isolated from whole blood collected from healthy pigs. Blood samples were mixed with lymphocyte separation medium (Solarbio, Beijing, China) and subjected to density gradient centrifugation at 1000× *g* for 30 min at room temperature using a horizontal centrifuge. RBCs were pelleted at the bottom of the tube, washed with D-Hanks’ solution (without Ca^2+^ and Mg^2+^), and stored in Alsever’s solution (Solarbio) at 4 °C.

The TGEV strain HN-2012 (Genbank: OP434397.1) was isolated by our laboratory. For virus infection assays, 10^7^ RBCs were incubated with TGEV at a multiplicity of infection (MOI) of 1. For the adhesion assay, RBCs were incubated with TGEV at 4 °C for 1 h. For the invasion assay, cells were incubated with TGEV at 37 °C for 1 h. For the time-course assays, cells were incubated with TGEV at 37 °C for 3, 6, 12, 24, 36, and 48 h. At each indicated time point, cells were washed with PBS to remove unbound virus and collected for analysis. Monolayer ST cells were also incubated with TGEV at 37 °C 1 h. Maintenance medium was then added for subsequent analyses.

### 2.2. Animals Challenge

Six 3-day-old healthy male piglets (Duroc × Landrace × Yorkshire) were obtained from a herd that had been confirmed to be free of TGEV, porcine epidemic diarrhea virus (PEDV), porcine deltacoronavirus (PDCoV), and porcine rotavirus (PoRV) by polymerase chain reaction (PCR) prior to purchase. All piglets were housed and cared for in accordance with the guidelines of the National Standard for Laboratory Animals and were allowed a two-day acclimation period to initiate the experiment. Piglets were randomly assigned to two groups: a control group (control) and a TGEV-infected group (TGEV). Animals were orally inoculated with 10 mL of DMEM (control) or TGEV (10^7^ TCID_50_) and housed in a separate room. Blood samples were collected using heparin as an anticoagulant at 72 h post-infection (hpi). Subsequently, all piglets were euthanized by intravenous injection of sodium pentobarbital (100 mg/kg body weight). Lung and small intestine samples were obtained and stored in liquid nitrogen or 4% paraformaldehyde.

### 2.3. Blood Collection and Analysis

Peripheral blood (2 mL) was collected from healthy or TGEV infected piglets into hemogram tubes with ethylenediaminetetraacetic acid (EDTA). Hematological parameters and plasma iron concentrations were measured with a BS-330E Automated Hematology Analyzer (Mindray, Shenzhen, China). Blood pH value and gas parameters were analyzed with an Abbott i-STAT System (Chicago, IL, USA).

### 2.4. RNA Extraction and Reverse Transcription Quantitative PCR (RT-qPCR) Assay

Total RNA was extracted from RBCs using FreeZol reagent (Vazyme, Nanjing, China). A total of 500 ng RNA was reverse transcription into cDNA using a HiScript II Reverse Transcriptase reagent kit (Vazyme) according to the manufacturer’s instructions. RT-qPCR was performed with a ChamQ Universal SYBR qPCR Master Mix kit (Vazyme). Specific primers targeting TGEV N gene were F: 5′-TGAAGGGCCAACGTAAAGAG-3′, R: 5′-CAACCCAGACAACTCC ATCTAA-3′.

### 2.5. Plaque Assay

The titer of TGEV in RBCs was determined by a plaque assay. TGEV was released from 10^7^ RBCs by freeze–thaw cycles in 400 μL DMEM. ST cells were seeded into 6-well plates, and 10-fold serial dilutions of TGEV (100 μL) were adsorbed onto monolayers at 37 °C for 1 h. Then cells were washed with PBS and overlaid with medium (containing DMEM, 2% FBS, and 1% low-melting-point agarose) for 72 h. Subsequently, cells were fixed and stained with 1% crystal violet solution prepared in methanol. Plaques were counted, and viral titers were expressed as plaque-forming units per mL (PFU/mL).

### 2.6. Flow Cytometry Analysis

The proportion of TGEV-positive RBCs was determined by flow cytometry. TGEV-infected RBCs were collected at the different time points post-infection. Cells were washed three times with PBS to remove unbound virus and then incubated with rabbit anti-TGEV polyclonal antibody (1:50, prepared by our laboratory) at 37 °C for 2 h. After washing three times with PBS, cells were stained with FITC-conjugated Goat anti-Rabbit IgG (1:100, Proteintech, Wuhan, China) for 30 min, washed again, and resuspended in 100 μL PBS. The percentage of TGEV-positive RBCs were analyzed using 10^5^ cells on a BD FACSCanto (Franklin Lakes, NJ, USA). Data were processed with FlowJo 10 software.

### 2.7. Prussian Blue Staining

Spleen tissues from piglets were collected and fixed for histological sectioning. After dewaxing and rehydration with xylene and gradient alcohol, the sections were incubated in Prussian blue solution for 30 min, rinsed with running water, and counterstained with eosin for 3 min. After dehydration and clearing, sections were mounted with gum for microscopic examination.

### 2.8. Diff-Quik Staining

To observe the morphology of RBCs, cells were suspended in PBS and smeared onto glass slide. After air drying, the smears were stained with a Diff-Quik staining kit (Solarbio) according to the manufacturer’s instructions.

### 2.9. Scanning Electron Microscope (SEM)

RBCs were fixed with 2.5% glutaraldehyde at 4 °C overnight. Samples were dehydrated through a graded ethanol serious, resuspended in anhydrous ethanol, mounted on tinfoil, and sputter-coated with gold. Images were acquired using an SEM (FEI).

### 2.10. Osmotic Fragility Test

Osmotic fragility was used to evaluate the integrity of the RBC membranes and detected by an Erythrocyte Osmotic Brittleness Test Kit (Solarbio). RBCs were incubated in NaCl solution of varying concentrations for 30 min. Membrane disruption led to hemolysis and release of hemoglobin, which was quantified by a measuring absorbance at 540 nm using a microplate reader (Thermos Scientific, Waltham, MA, USA). The hemoglobin concentration reflected the percentage of RBC lysis.

### 2.11. Phosphatidylserine (PS) Measurement

Externalization of PS on the RBC membranes was assessed by an Annexin V-FITC kit (Beyotime, Beijing, China). RBCs were resuspended in Annexin V-FITC binding buffer at a density of 1 × 10^6^ cells/mL. Annexin V-FITC (5 μL) was added to the suspension and incubated for 20 min at room temperature. Cells were then washed with PBS and analyzed by flow cytometry (BD).

### 2.12. Raman Spectroscopy Analysis

Hemoglobin content in RBCs was quantified using Raman spectroscopy (HORIBA Paris, France). Single RBCs suspended in PBS were irradiated with a 532 nm laser at a maximum power of 100 mW. A 100× objective lens was used to collect spectra within the range of 300–2000 cm^−1^. Each sample was scanned with an exposure time of 15 s, repeated for two cycles. For each experimental group, spectra were collected from at least 10 individual cells. Peak assignments of the Raman spectra are summarized in Table 1.

### 2.13. Detection of Oxyhemoglobin and Methemoglobin

The levels of oxyhemoglobin and methemoglobin in RBCs (1 × 10^7^ cells) were detected by a Methemoglobin content assay kit (Solarbio). Briefly, RBCs were washed with PBS and detected at the absorbance of 560 nm and 630 nm by a spectrophotometer.

### 2.14. Western Blot (WB) Analysis

Total proteins were extracted, mixed with loading buffer, and boiled. Cellular proteins were separated by 12% sodium dodecyl sulfate–polyacrylamide gel electrophoresis and transferred to PVDF membranes. The membranes were blocked with 5% non-fat milk for 2 h and then incubated with monoclonal antibody against TGEV N protein (1:3000, prepared by our laboratory) or Anti-Transferrin Receptor antibody (1:1000, Abcam, Cambridge, UK) at 4 °C overnight. After three washes with TBST, membranes were incubated with HRP-conjugated Goat Anti-Mouse IgG or HRP-conjugated Goat Anti-Rabbit IgG (1:5000, Proteintech) for 2 h. Proteins were visualized using an enhanced chemiluminescence reagent (Beyotime) and imaged using an AmershamImager800 system.

### 2.15. Statistical Analysis

Data are expressed as the means ± standard deviations (SDs) from three independent experiments. All data in this study were analyzed using one-way ANOVA by SPSS version 26.0 software. Different letters indicate significant difference (*p* < 0.05), whereas identical letters indicate no significant difference (*p* > 0.05).

## 3. Results

### 3.1. Comparison of Hematological Parameters Between Healthy and TGEV-Infected Piglets

To confirm TGEV had established a successful infection, 5-day-old piglets were challenged with TGEV. Lung and small intestine samples were collected, and total RNA was extracted and subjected to RT-qPCR. As shown in Figure 1, the limit of detection for TGEV was approximately 10^3.8^ copies/mL. Viral loads in the lungs were significantly higher than those in the small intestine.

Since TGEV-infected piglets exhibited symptoms of hypoxia, the effects of TGEV infection on RBCs were firstly evaluated by hematology parameters. Blood specimens from TGEV-infected and healthy piglets were compared (Table 2). Both RBC distribution width–coefficient of variation (RDW_CV) and RBC distribution width–standard deviation (RDW_SD) were significantly lower in TGEV-infected piglets than healthy controls. Other parameters, including RBC count, hemoglobin (HGB), hematocrit (HCT), mean corpuscular volume (MCV), platelet count (PLT), mean platelet volume (MPV), platelet distribution width (PDW), and plateletcrit (PCT), were also lower in TGEV-infected piglets, although the difference were not statistically significant. Given that RBCs are responsible for transporting oxygen and carbon dioxide, blood pH and blood gas levels were further analyzed. The pCO_2_ was markedly elevated in TGEV-infected piglets, while blood pH and pO_2_ remained unaffected.

### 3.2. Detection of TGEV RNA in RBCs from Infected Piglets

To assess whether TGEV can infect RBCs, blood samples were collected from TGEV-infected piglets. High levels of TGEV RNA were detected in RBCs from one of the three piglets (Figure 2A). No plaque was observed in samples from any piglet, except for TGEV-positive samples, where plaque formation was evident (Figure 2B). Flow cytometry confirmed a low percentage of TGEV-infected RBCs (Figure 2C,D). Together, these results indicated that TGEV can infect RBCs.

### 3.3. TGEV Binds to RBCs

To investigate the infection characteristics of TGEV in RBCs, RBCs from healthy piglets were isolated, incubated with TGEV, and collected at different time points post-infection. RT-qPCR analysis showed that TGEV mRNA levels gradually decreased over time (Figure 3A). The plaque assay indicated that viral titers were undetectable at 12 hpi (Figure 3B,C). Flow cytometry further validated that the percentage of TGEV-positive cells continuously declined and was no longer detectable at 12 hpi (Figure 3D,E). Considering the first replication cycle of coronavirus is approximately 6 h, these findings suggested that TGEV can bind to, but not replicate in, RBCs [8,9].

Sialic acid (SA), a major glycocalyx carbohydrate on RBCs, has been reported to function as an adhesion receptor during TGEV infection. To assess the role of SA in TGEV binding, RBCs were pretreated with 0.1 U/mL neuraminidase (NA) prior to TGEV inoculation. RT-qPCR revealed that TGEV mRNA expression in NA-treated RBCs was significantly reduced at both the adhesion and invasion stages (Figure 4A). Consistent with the RT-qPCR results, virus titers (Figure 4B,C) and the proportion of TGEV-positive cells (Figure 4D,E) were also decreased after NA pretreatment. These results demonstrated that SA is an important mediator of TGEV binding to RBCs.

### 3.4. TGEV Binding Results in Structural Damage to RBCs

Because viral infection can cause damage to the RBCs, the effects of TGEV binding on RBC morphology, deformability, and osmotic fragility were examined. Diff-Quik staining revealed that normal RBCs were round and smooth, whereas TGEV-exposed RBCs appeared wrinkled, crenated, and displayed membrane extensions (Figure 5A). Flow cytometry analysis showed that an increase in side scatter (SSC), which reflects cellular complexity, suggesting membrane protrusions. In contrast, forward scatter (FSC), which reflects cell size, was decreased, likely due to cellular shrinkage (Figure 5B). SEM images further demonstrated that normal RBCs had a diameter of 5 to 7 μm, while TGEV-infected RBCs displayed numerous protrusions characteristic of “echinocytes” (Figure 5C).

Osmotic fragility, an important indicator of RBC membrane stability, was significantly increased with both prolonged infection and higher infection doses (Figure 6A,B), suggesting reduced deformability of TGEV-infected RBCs. Additionally, exposure of PS on the RBC surface was markedly elevated following TGEV infection and increased with time (Figure 6C,D). These findings collectively demonstrated that TGEV infection disrupts RBC membrane integrity.

### 3.5. TGEV Decreases the Oxygen-Carrying Capacity of RBCs

The structure integrity of RBCs is closely linked to their oxygen transport function. To assess the effect of TGEV infection on RBC function, oxygen contents in RBCs were analyzed using Raman spectroscopy. The average Raman spectra of control and TGEV-infected RBCs are shown in Figure 7A. The 1225 cm^−1^ band, corresponding to the bending vibration of the C-H methyl plane and known to weaken under hypoxic conditions, was reduced in TGEV-infected cells. The bands at 1355 cm^−1^ and 1375 cm^−1^ serve as reference indicators to distinguish hypoxia from oxygen-rich states. The peak at 1639 cm^−1^, which reflects high oxygen content, was also diminished in TGEV-infected RBCs. Moreover, the 1375 cm^−1^ peak shifted to 1355 cm^−1^ after infection. The ratio of Raman intensity at 1375 cm^−1^/1355 cm^−1^, representing the relative abundance of hemoglobin and ligand complexes (oxyhemoglobin), was significantly decreased (Figure 7B). Oxyhemoglobin levels were further measured by a methemoglobin assay. As shown in Figure 7C, oxyhemoglobin progressively declined with prolonged TGEV infection. In contrast, methemoglobin, which is incapable of oxygen transport, was increased (Figure 7D). These results demonstrated that TGEV infection reduces the oxyhemoglobin content and compromises the oxygen-carrying function of piglet RBCs.

### 3.6. TGEV Binding Enhances Macrophage Phagocytosis of RBCs

Aging or structurally damaged RBCs are normally cleared by macrophages. To determine whether TGEV infection influences erythrophagocytosis, macrophages were co-cultured with TGEV-pretreated RBCs. Macrophage SSC values increased after exposure to RBCs and further elevated when TGEV-infected RBCs were added (Figure 8A). When RBCs were labeled with a live-cell fluorescent dye and co-cultured with macrophages, the fluorescence intensity of macrophages engulfing TGEV-infected RBCs was stronger than that of macrophages exposed to uninfected RBCs (Figure 8B,C). These findings indicated that TGEV infection enhances macrophage phagocytosis of RBCs.

### 3.7. Iron Influence TGEV Infection

Normal iron homeostasis is disrupted during viral infections. Hemosiderin, an iron-storage complex formed after macrophages engulf RBCs, can be detected by Prussian blue staining. As shown in Figure 9A, abundant blue granules of hemosiderin were observed in the spleen and liver of TGEV-infected piglets, indicating iron was “locked” in macrophages, which may limit iron export and contribute to low plasma iron. Consistent with this observation, serum iron levels in TGEV-infected piglets were significantly decreased (Figure 9B). Transferrin receptor 1 (TFR1) has been demonstrated to serve as the invasion receptor for TGEV [10]. To further assess the effect of the iron concentration on viral infection, ST cells were pretreated with the iron chelator deferoxamine mesylate (DFOM) to limit iron availability or with ferric ammonium citrate (FAC) to supplement iron and then infected with TGEV. DFOM treatment enhanced the expression of TFR1, while FAC treatment inhibited its expression (Figure 9C). TGEV mRNA and protein expression were increased in DFOM-treated cells, whereas FAC decreased viral infection (Figure 9D,E). These results suggested that erythrophagocytosis-mediated alterations in iron levels can modulate TGEV infection.

## 4. Discussion

TGEV is a major swine coronavirus that causes fatal diarrhea in piglets less than two weeks old. The prevalence of TGEV has decreased over the decades due to vaccination. However, variant TGEV strains have been reported in US and China. Large amounts of virions are present in the small intestine and lungs [4]. The virulence, tropism, and pathogenicity of these recombinant TGEV strains have evolved, complicating the prevention and control of TGEV. During our experiment, breathlessness was observed in TGEV-infected piglets, which was suspected to be related to altered oxygen uptake by RBCs. Hypoxia, a hallmark of SARS-CoV-2 pneumonia, has prompted increased interest in the interactions between viruses and RBCs. Therefore, effects of TGEV infection on RBCs and potential pathophysiological roles in disease progression were explored in this study.

As obligate intracellular parasites, viruses rely on host cellular machinery for genome replication and protein synthesis. Because mature RBCs in mammals lack nuclei and other organelles, their involvement in viral infections is often overlooked. However, studies have shown that RBCs can act as temporary “transport vehicles”, facilitating the spread of viruses through the bloodstream [7]. Our results demonstrated that TGEV could bind to RBCs, suggesting TGEV has the ability to infect RBCs. Clinically, the presence of viral RNA in the blood is termed viremia. Detection of viral RNA in plasma samples has been observed in 75% of SARS-CoV-2 and 90% of PEDV infections, respectively [11,12]. Furthermore, PDCoV has been identified in blood samples from piglets and Haitian children [13,14,15]. Since coronavirus can be detected in multiple organs, viremia may contribute to the distant transmission of the virus, leading to broad tissue tropism [16,17]. Additionally, viremia is associated with disease severity and high mortality [18,19]. Collectively, these findings enhance our understanding of TGEV infection and transmission, highlighting the importance of diagnostic technologies targeting RBCs.

Viral infections are initiated by attachment to host cell receptors. In the case of RBCs, viruses bind primarily to the plasma membrane, transmembrane proteins, or glycocalyx carbohydrates [7]. Although specific viral receptors are not expressed on RBCs, SA has been shown to interact with the spike (S) protein of coronavirus. TGEV binds to SAs through the N-terminal domain (NTD) of its S protein [20,21]. SARS-CoV-2 attaches to SA primarily through glycans at 22 N-glycosylation sites on the viral S1 protein [22]. Pretreatment of RBCs with NA reduced partial TGEV adhesion, suggesting other components of the RBC membrane might contribute to TGEV infection. Band 3, CD147, and β1 chains of hemoglobin have been reported as potential binding receptors for SARS-CoV-2 infecting RBCs [23,24]. Given RBCs’ ability to express viral receptors and permeate all organs, they can be selectively coated with viral receptors to minimize the invasion of infectious particles into susceptible tissues or loaded with drugs to eliminate virus-infected cells [25,26]. These findings offer new insights into the prevention and control of TGEV, and the precise mechanism by which TGEV binds to RBCs warrants further investigation.

Abnormalities in RBCs are commonly observed in many infectious diseases. The frequency of diverse RBC morphology may serve as a predictor of disease severity during virus infection. In TGEV-infected RBCs, serrated, elongated, and crenated morphologies were observed likely due to defects in RBC membrane composition. Thomas et al. combined multi-omics approaches to investigate the impact of SARS-CoV-2 infection on RBC membrane composition. Proteomics analysis revealed that ankyrin, spectrin β, and band 3, which comprise the response maintaining RBC membrane integrity, underwent oxidation and fragmentation. Lipidomics data indicated viral infection significantly reduced the synthesis of lipid classes, altering membrane lipids remodeling pathways. Metabolomics showed an increase in glycolytic activity in SARS-CoV-2-infected RBCs, which may theoretically enhance hemoglobin’s oxygen off-loading to counteract hypoxia [27,28].

Dysfunctional RBCs result in a limited oxygen supply, contributing to hypoxia and associated complications. Hemoglobin, the primary oxygen carrier, is responsible for approximately 98% of total oxygen transport. TGEV infection reduced oxygen supply by binding to RBCs, potentially causing hemoglobin damage. Wiedemann et al. found that SARS-CoV-2 can directly infect RBC progenitors, impairing hemoglobin synthesis during RBC maturation and turnover [29]. Liu and Lechuga et al. further demonstrated that multiple SARS-CoV-2 proteins act as potential ligands binding to hemoglobin, inducing hemoglobin denaturation and dysfunction, which in turn impairs oxygen uptake, binding, or release [30]. In TGEV-infected piglets, altered hematological parameters were consistent with impaired oxygen delivery. However, due to limitations in the number of animals, hematological changes in TGEV-infected piglets should be further investigated. Numerous clinical studies have indicated that COVID-19 is associated with changes in hematological parameters, which are more pronounced in patients with severe diseases [31]. Notably, abnormal hematological parameters and morphological changes were observed even after 60 days’ recovery from mild COVID-19, suggesting that coronavirus may cause long-term effects in hosts [32]. Together, these results demonstrated that TGEV infection reduced oxygen supply of RBCs and may cause other symptoms such as anemia.

Currently, there are no reports describing TGEV-induced RBC damage or its underlying mechanisms. Based on the literature, we speculate that there are potential mechanisms by which TGEV affects RBCs. TGEV binds to RBCs via membrane-associated receptors, subsequently altering the cytoskeleton beneath the plasma membrane, leading to morphological abnormalities and disrupting hemoglobin, thereby impairing RBC function.

Old or diseased RBCs are typically engulfed by macrophages in the red spleen or liver. Two factors contribute to macrophage-mediated RBC clearance. First, TGEV infection reduces RBC deformability, preventing the cells from passing through inter-endothelial slits and capillaries narrower than the RBC diameter. Second, several pro-phagocytic signals are generated to recruit macrophages. PS, normally located on the inner leaflet of the healthy RBC membrane, can act as an “eat me” signal when exposed on the outer leaflet under some pathological conditions. Additionally, membrane protein CD47 interacts with signal-regulatory protein alpha to inhibit phagocytosis. Clustering and conformational changes in band 3 are recognized by natural antibodies, leading to the formation of immune complexes with complement C3, which triggers phagocytosis [7,33,34]. These phagocytic signals are essential for RBC clearance. Enhanced PS exposure in TGEV-infected RBCs was consistent with increased macrophage-mediated RBC engulfment. Although macrophages are immune cells that defense against viral invasion, erythrophagocytosis can capture viruses, providing an alternative route facilitating viral entry into the cells [35]. African swine fever virus (ASFV) has been shown to adsorb to RBCs, inducing apoptosis and enhancing macrophage phagocytosis, which in turn increases the efficiency of ASFV entry [36]. Currently, vaccines utilizing RBCs to deliver antigens to macrophages are being developed to induce specific cellular immunity.

Macrophages scavenge RBCs to release iron into the circulatory system. Iron is crucial for developing erythroblasts and is also an essential nutrient for microbial infection. In the host, absorbed iron is bound to transferrin, which interacts with its receptor, TFR1, on the cell surface for internalization. Recent studies have shown that iron chelator DFOM or iron supplements like FAC can alter iron balance and affect viral infection [37]. TFR1 has also been confirmed to act as a receptor mediating various virus invasion, including SARS-CoV-2, IAV, PEDV, and TGEV [10,38,39,40]. Clinically, newborn piglets are highly susceptible to PEDV infection, partly due to iron deficiency and the high expression levels of TFR1, which facilitate viral infection [40]. Our results showed that iron levels in TGEV-infected piglets were deposited in the spleen as hemosiderin, while plasma iron levels were reduced, potentially favoring viral infection through enhanced TFR1 expression in intestinal epithelial cells. Low serum iron had been reported to contribute to dengue virus, SARS-CoV-2, and HIV infection [29,41,42]. Serum iron levels in severe COVID-19 patients were significantly lower than those in mild COVID-19, suggesting a strong correlation between low serum iron and disease severity. Thus, providing an appropriate amount of iron may be crucial for defense against pathogen infections.

## 5. Conclusions

In summary, our study demonstrates that TGEV infection results in injury to RBCs and induces hypoxia in piglets, which plays a significant role in the disease progression of viral infection. Our findings underscore the potential of diagnostic technologies targeting RBCs and suggest new avenues for drug development and vaccine design for TGEV prevention. Further exploration of the mechanisms by which RBCs contribute to viral infection is warranted.

## Figures and Tables

**Figure 1 vetsci-13-00042-f001:**
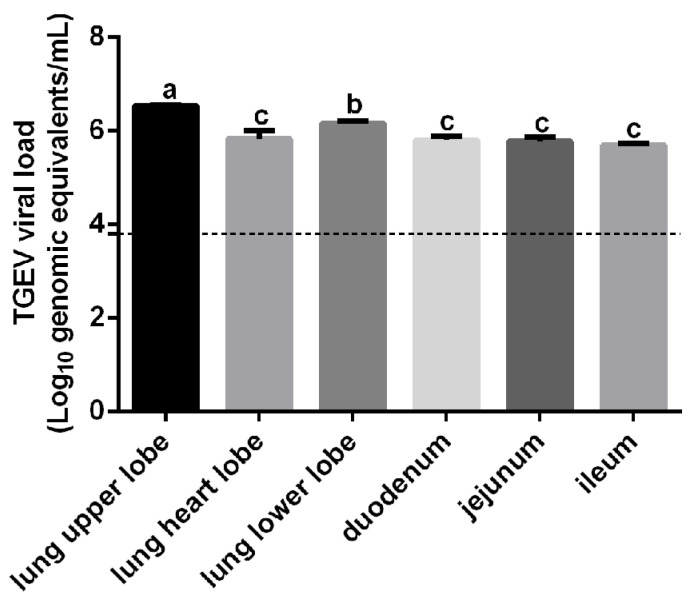
Detection of TGEV in the lung and small intestine. Viral loads in collected samples were quantified by RT-qPCR. Different letters indicate significant difference (*p* < 0.05), whereas identical letters indicate no significant difference (*p* > 0.05).

**Figure 2 vetsci-13-00042-f002:**
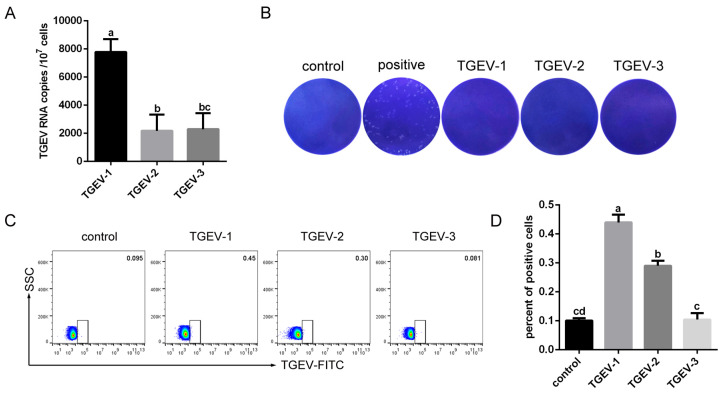
Detection of TGEV in RBCs from infected piglets. (**A**) Viral RNA levels in RBCs were quantified by RT-qPCR. (**B**) Infectious viral titers in RBCs were determined by the plaque assay. (**C**) TGEV-positive RBCs were identified by flow cytometry. (**D**) Quantification of TGEV-positive cells. Different letters indicate significant difference (*p* < 0.05), whereas identical letters indicate no significant difference (*p* > 0.05).

**Figure 3 vetsci-13-00042-f003:**
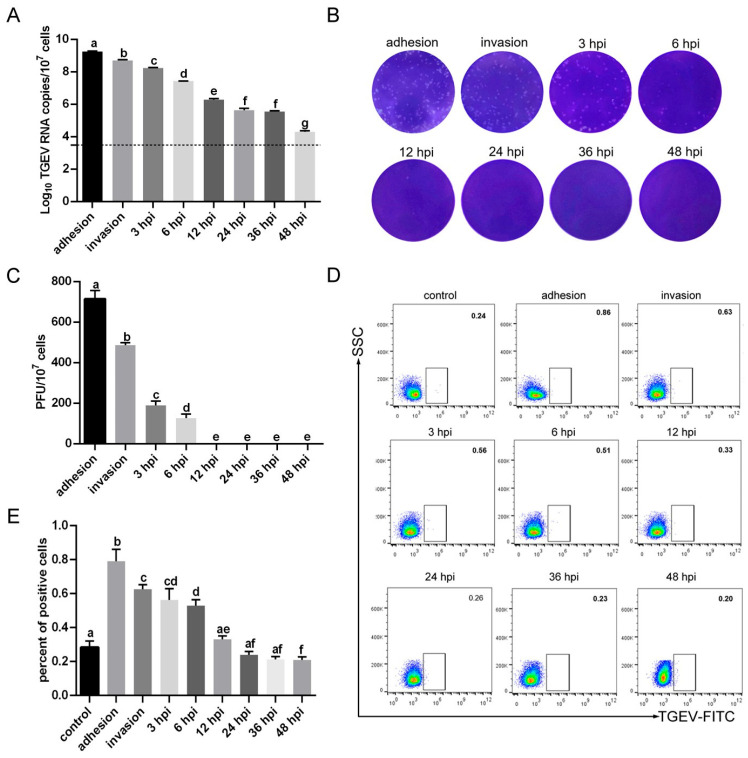
TGEV binding to RBCs. (**A**) TGEV RNA copies were quantified by RT-qPCR. (**B**) Viral titer in RBCs was determined by the plaque assay. (**C**) Quantification of viral titer. (**D**) Percentages of TGEV-positive RBCs were detected by flow cytometry. (**E**) Quantification of TGEV-positive RBCs. Different letters indicate significant differences (*p* < 0.05), whereas identical letters indicate no significant difference (*p* > 0.05).

**Figure 4 vetsci-13-00042-f004:**
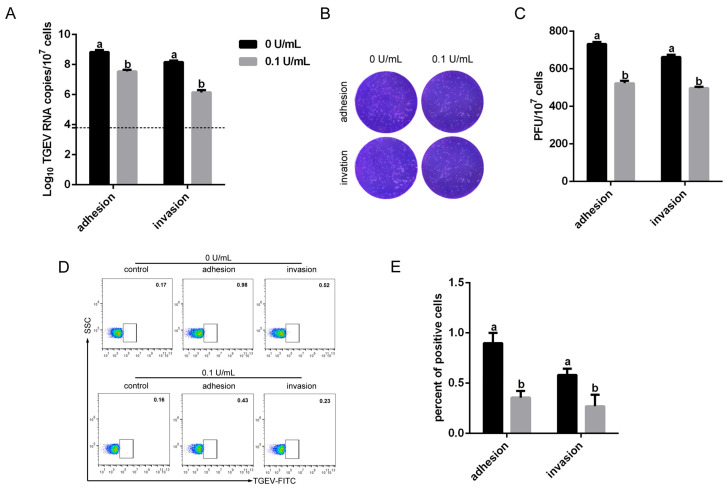
Role of SA in TGEV binding to RBCs. RBCs were pretreated with 0.1 U/mL NA at 37 °C for 1 h and subsequently inoculated with TGEV at 4 °C or 37 °C for 1 h. Viral RNA levels, infectious titers, and percentages of TGEV-positive RBCs were measured by RT-qPCR (**A**), the plaque assay (**B**,**C**), and flow cytometry (**D**,**E**), respectively. Different letters indicate significant differences (*p* < 0.05), whereas identical letters indicate no significant difference (*p* > 0.05).

**Figure 5 vetsci-13-00042-f005:**
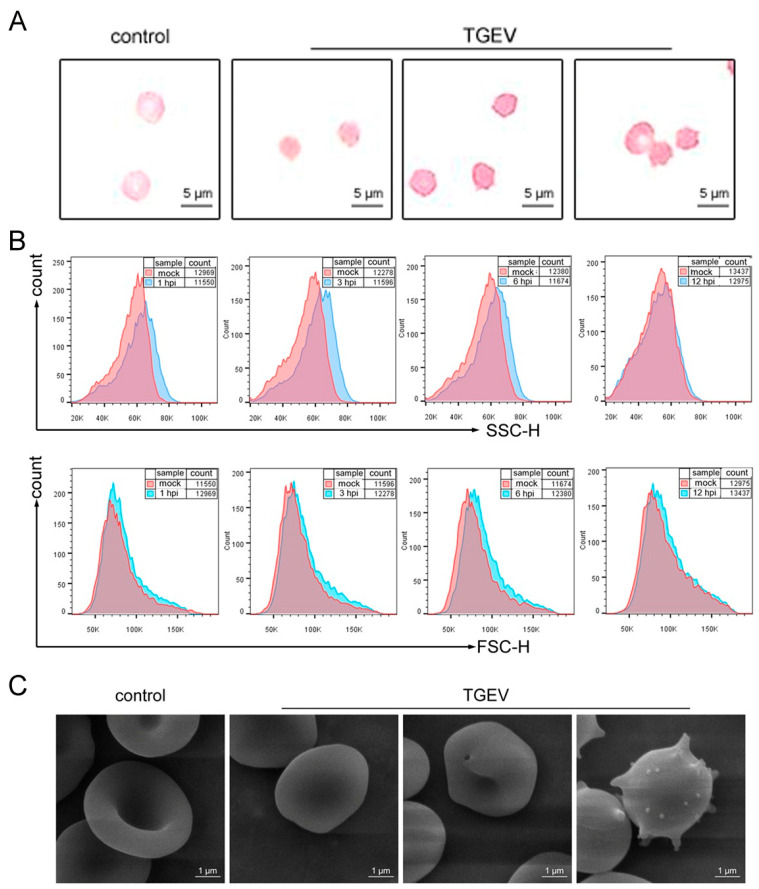
Morphology abnormalities of TGEV-infected RBCs. RBCs were inoculated with TGEV and analyzed by multiple approaches. (**A**) Diff-Quik staining images were obtained at ×40 magnification. Scale bars = 5 μm. (**B**) FSC and SSC parameters of RBCs were analyzed by flow cytometry. (**C**) SEM image showing ultrastructural changs in RBC. Scale bars = 1 μm.

**Figure 6 vetsci-13-00042-f006:**
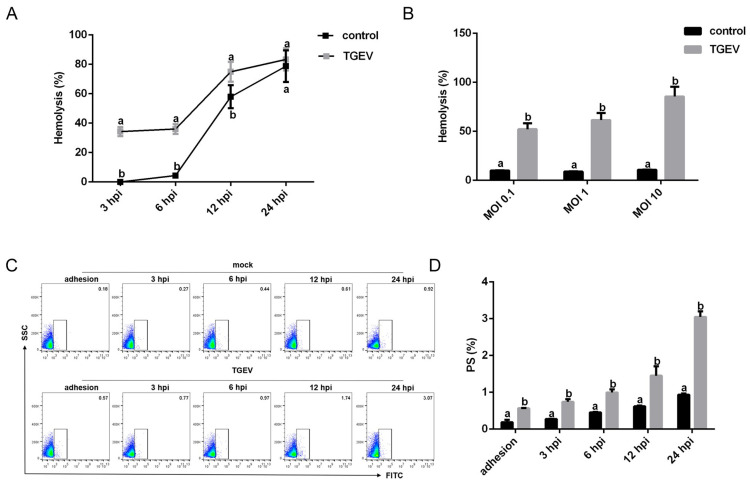
Effects of TGEV adhesion on RBC membrane integrity. (**A**) Osmotic fragility of RBCs infected with TGEV (MOI = 1) were detected at 3, 6, 12, and 24 hpi. (**B**) Osmotic fragility of RBCs was measured after incubation with TGEV at different MOIs for 1 h. (**C**) PS exposure on RBCs was detected by flow cytometry. (**D**) Quantification of PS-positive RBCs. Different letters indicate significant differences (*p* < 0.05); the same letter indicates no significant difference (*p* > 0.05).

**Figure 7 vetsci-13-00042-f007:**
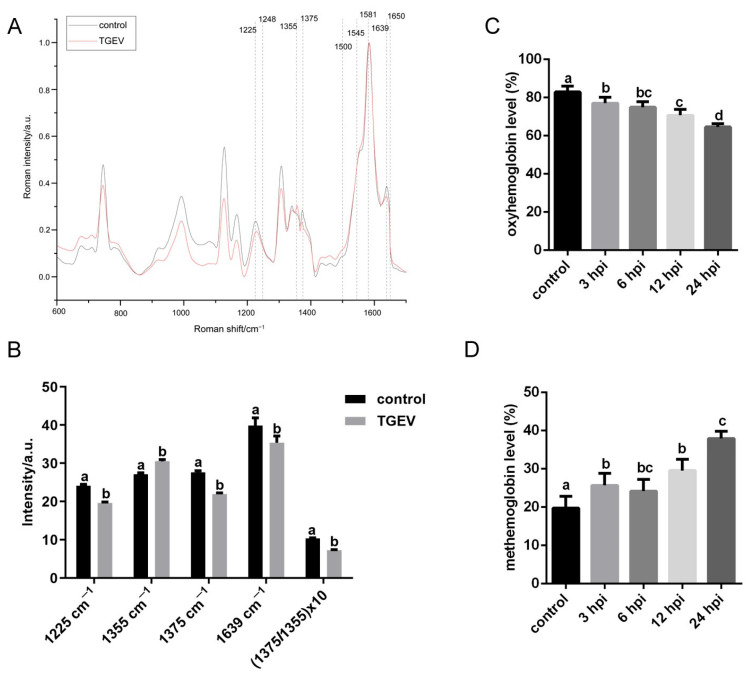
Effects of TGEV infection on hemoglobin in RBCs. (**A**) Average Raman spectra of RBCs. (**B**) Histogram of the most significant Raman peaks. (**C**,**D**) Levels of oxyhemoglobin (**C**) and methemoglobin (**D**) in TGEV-bound RBCs measured at 3, 6, 12, and 24 hpi. Different letters indicate significant differences (*p* < 0.05); the same letter indicates no significant difference (*p* > 0.05).

**Figure 8 vetsci-13-00042-f008:**
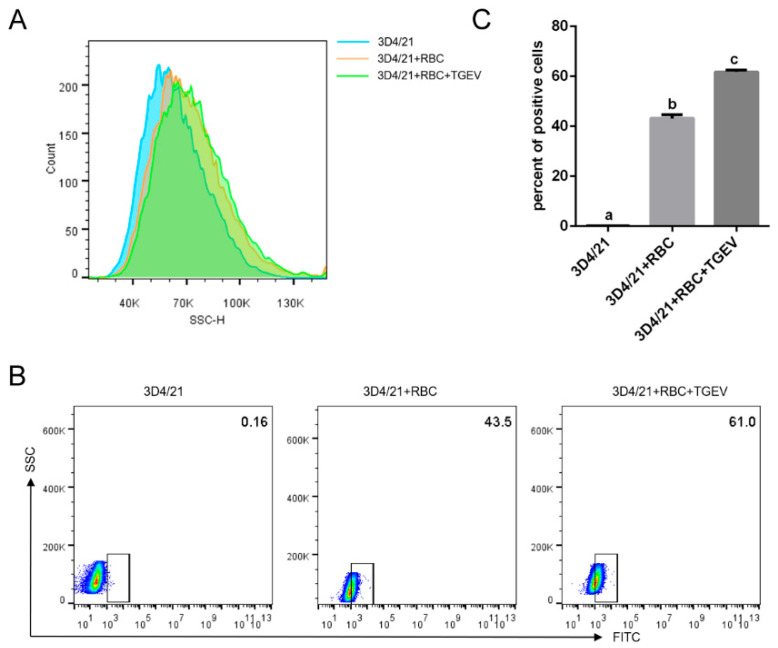
Effects of TGEV infection on macrophage phagocytosis. (**A**) SSC values of macrophages were analyzed by flow cytometry. (**B**) Fluorescence intensity of macrophages engulfing RBCs was measured by flow cytometry. (**C**) Quantification of fluorescence intensity. Different letters indicate significant differences (*p* < 0.05); the same letter indicates no significant difference (*p* > 0.05).

**Figure 9 vetsci-13-00042-f009:**
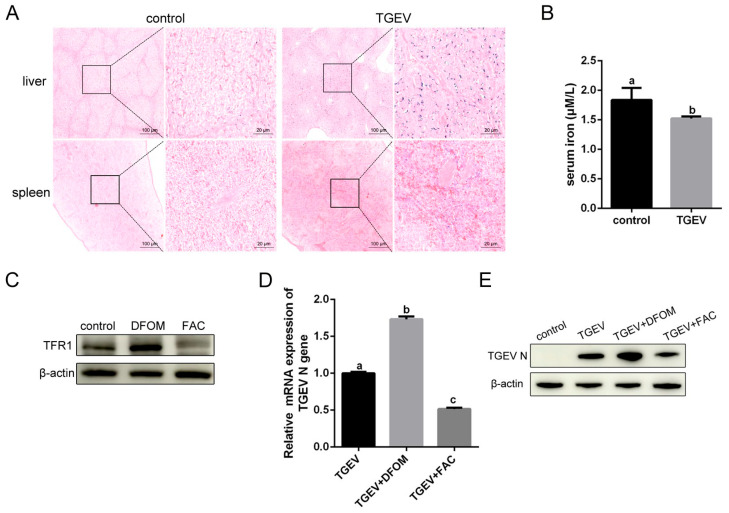
Iron affects TGEV infection. (**A**) Microscopic observations of liver and spleen sections stained with Prussian blue. Scale bars = 20 or 100 μm. (**B**) Serun iron in TGEV-infected and healthy piglets. (**C**) ST cells were pretreated with DFOM (100 μM) or FAC (100 nM) for 6 h to detect the expression of TFR1. (**D**,**E**) ST cells were pretreated with DFOM or FAC and then infected with TGEV (MOI = 1). Viral RNA or viral protein was analyzed by RT-qPCR (**D**) or WB (**E**) at 24 hpi. Different letters indicate significant differences (*p* < 0.05); the same letter indicates no significant difference (*p* > 0.05).

**Table 1 vetsci-13-00042-t001:** Raman spectrum peak position and material identification.

Peak Position (cm^−1^)	Assignment
1225	protein
1248	Amide III/heme aggregation
1355	Fe^2+^/deoxyHb
1375	Fe^3+^/oxyHb
1565	tryptophan
1585	cytochrome c
1639	hemoglobin
1650	Amide I

**Table 2 vetsci-13-00042-t002:** Blood parameters of the piglets.

Blood Parameters	Unit	Control	TGEV
RBC	10^12^/L	6.477 ± 3.059	4.277 ± 1.767
HGB	g/L	116.3 ± 51.87	78.33 ± 32.72
HCT	%	38.43 ± 17.25	24.27 ± 9.765
MCV	fL	59.97 ± 1.234	57.13 ± 5.297
MCH	pg	18.1 ± 0.4359	18.37 ± 2.996
MCHC	g/L	302.3 ± 3.215	321 ± 21.63
RDW_CV	%	27.7 ± 3.503 ^a^	19.73 ± 1.358 ^b^
RDW_SD	fL	50.57 ± 2.829 ^a^	38.13 ± 2.779 ^b^
PLT	10^9^/L	889 ± 426.9	596 ± 369.9
MPV	fL	12.17 ± 3.753	10 ± 0.7
PDW	%	14.93 ± 0.5033	14.63 ± 0.6658
PCT	%	0.2473 ± 0.1426	0.1603 ± 0.007506
pH		7.265 ± 0.1526	7.237 ± 0.1617
O_2_	mmHg	51.33 ± 8.622	50.67 ± 4.041
CO_2_	mmHg	35.33 ± 2.887 ^a^	42.33 ± 2.887 ^b^

Note: HGB: hemoglobin; HCT: hematocrit; MCV: mean corpuscular volume; MCH: mean corpuscular hemoglobin; MCHC: mean corpuscular hemoglobin concentration; RDW_CV: red blood cell distribution width–coefficient of variation; RDW_SD: red cell distribution width–standard deviation; PLT: platelet count; MPV: mean platelet volume; PDW: platelet distribution width; PCT: plateletcrit. Different letters indicate significant difference (*p* < 0.05), whereas identical letters indicate no significant difference (*p* > 0.05).

## Data Availability

The original contributions presented in this study are included in the article/Appendix A. Further inquiries can be directed to the corresponding authors.

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
