# Peer review of "Transmissible Gastroenteritis Virus Binding to Red Blood Cells Disrupts Iron Homeostasis and Promotes Viral Infection"

_vetsci, 2026, doi:10.3390/vetsci13010042_

Round 1
Reviewer 1 Report
Comments and Suggestions for Authors
In this manuscript, the authors demonstrated that TGEV infection pose significant effects to RBC. TGEV could bind to, but not replicate in RBCs and furtherpromoted macrophage-medicated phagocytosis of RBCs, leading to decreased serum iron levels. The results presented in the manuscript is full of contradiction and conflict. There are several issuers needed to be addressed.
Major comments:
- The specific binding of TGEV to RBCs has not been presented in the manuscript. Although the flow cytometry and qPCR analysis provided data, the key evidence has not been provided. The result of immunoelectron microscopy observation must been included.
- 1, TGEV mainly targets the intestine but not the lung. However, the viral loading is nearly the same among the intestine and lung. Why? Has PRCV strain been used in this study? The limit of detection (10 3.8 copies/mL) is so high and the error bar is very small, which indicating that the result is unreliable. Please explain the puzzled result.
- The blood specimens changed during TGEV infection is a common phenomenon because TGEV infection leads to serious diarrhea and dehydration, which greatly contribute to blood specimens. Thus, this is not a evidence for TGEV infection in RBCs.
- The control is missing in Fig.2A. Additionally, as indicated by qPCR, the results are highly below the detection limition (10 8 copies/mL). Are the results reliable?
- 4B, the plaque result of 0.1U/mL is the same. Are the results credible?
- 5, no results showed TGEV virions binding to RBCs. The ultrastructural changes may caused by various factors.
- 8, key control is missing. 3D4/21 should also mixed to TGEV.
- How to connected the results of “Iron Inffuence TGEV Infection”to RBCs? TGEV is binding to so few cells. How the influences be enlarged in liver and spleen?

Reviewer 2 Report
Comments and Suggestions for Authors
The manuscript by Xia et al. investigated the pathophysiological implications of transmissible gastroenteritis virus (TGEV) binding to red blood cells (RBCs). TGEV infection causes severe damage to the lining of the small intestine, resulting in Gastroenteritis, which is a highly contagious viral disease. The TGEV spike protein has a specific domain that binds to sialic acid on the surface of red blood cells, a property known as hemagglutination, which is considered a pathogenicity factor and used for diagnoses. This study is well-structured, and the findings revealed that TGEV binds to RBCs and causes the RBCs to become deformed and less flexible, leading to a reduced capacity to carry oxygen. The deformed RBCs are more likely to be engulfed and destroyed by macrophages, which can potentially lead to lower serum iron levels, creating an environment that enhances viral infection. This provides a plausible explanation for why infected animals might show systemic illness and weakness beyond gastrointestinal symptoms. The major limitation is the small sample size in the animal study, with only 3 piglets per group. In addition, the following concerns need to be addressed.
- From lines 89 to 92, the timeline of sampling is not clear; the blood sampling should be collected before euthanization.
- Section 2.3, titled “Blood collection and sampling,” is not accurate. This section is more about the blood analysis instead of sampling.
- Section 2.4 title is not correct; the method used in the study is RT-qPCR, not qPCR. The same applies to other sections. In addition, you should describe the detailed technical information, such as the number of RBCs used for RNA extraction and the amount of RNA used in RT and qPCR conditions, or provide a reference for it.
- Lines 107 to 113: The number of RBCs used for virus titration, the detailed method of virus release, the dilution range, the volume of inocula, etc., should be described.
- Section 2.6. While the general steps are outlined, crucial procedural parameters and reagent information are missing. This section requires significant expansion and clarification.
- Section 2.14 lacks technical details, such as the specificity and source of antibodies, etc.
- In section 2.15, the statistical description is incomplete and fails to specify the test(s) actually performed.
- In section 3.1. Comparison of Hematological Parameters Between Healthy and TGEV-Infected Piglets. There is no description of the sample preparation here or methods section; the amount of each sample has to be comparable. "Per mL" is a vague description.
Reviewer 3 Report
Comments and Suggestions for Authors
In this manuscript the authors investigate the transmissible gastroenteritis virus (TGEV), a swine coronavirus which causes diarrhea and high mortality in piglets. The authors, in an in vivo assay, observed that TGEV infection causes an alteration in hematological parameters, impairing oxygen transportation. They observed that this virus can bind to red blood cells (RBCs) mediated by sialic acid residues but cannot replicate within them. This binding induced structural damage to RBCs, decreasing the capacity of oxygen-carrying and making them more readily to macrophage-mediated phagocytosis. All these features led to a decreasing in oxyhemoglobin, decreasing in metahemoglobin, iron deposits in spleen and liver and reducing of iron levels. Interestingly, the authors also observed in vitro that iron supplementation enhances TGEV infection. All these observations will be useful to design new diagnostic and therapeutic strategies against this disease.
The topic is novel, and the results are quite interesting. Nevertheless, there are some points which need to be arranged prior to considering this manuscript to be accepted.
-The main concern about this manuscript is the low number of animals in the experimental design (n=3 per group). Such a low number limits the impact of the statistical results and the results obtained. Therefore, the authors may justify the use of only 6 animals in this report or discuss this limitation in the Discussion section.
- Regarding the RBCs extraction, did the authors check the purity of the extraction method? Additionally, did they discard the possibly contamination of nuclear cells contamination?
- In lines 91-92 specify the authors used heparin to collect blood samples, but in lines 95-96 they specify that they also used EDTA as an anticoagulant to collect blood samples. Are these methods to blood sampling related to different essays? Please, specify.
- The statistical analysis methodology is incomplete. Please, specify what statistical tests were used and the normality test employed.
- Figures: the letters used to reflect the statistical differences in some pannels are a bit confusing. I.E.: Fig. 3, panel A, the significance (a,b,c,d…) is compared to what group (or groups)? This matter should be arranged to clarify.
Reviewer 4 Report
Comments and Suggestions for Authors
TGEV is an enteric coronavirus that responsible for vomiting, diarrhea and dehydration in piglets less than two weeks old. In the manuscript, it was demonstrated that TGEV possesses the capability to bind to RBCs but does not replicate within them. This binding lead to the disruption of both the structure and function of RBCs. Consequently, macrophage-mediated phagocytosis of RBCs is promoted, resulting in decreased serum iron levels. These factors, in turn, enhance TGEV infection. The authors delved into the pathogenesis of TGEV infection from a pathophysiological perspective, thereby providing novel insights that hold promise for the development of diagnostic and therapeutic strategies. However, there are some suggestions that require the authors to make further improvements. Upon satisfactory revision, the work may be suitable for publication in Veterinary Sciences. Here are some specific comments for revisions:
- Line 22, “macrophage-medicated”should be changed to “macrophage-mediated”.
- Lines 42-44, it is recommended to clarify the background of the clinical observations. Whether these observations were derived from TGEV animal challenge experiments or from clinical cases during pig farm breeding? Clinically, dyspnea in piglets is often associated with co-infections of other pathogens (e.g., PRRSV, SIV) rather than being a specific symptom of TGEV infection alone.
3.Given that TRF1 has been established to influence virus infection, it is necessary to confirm whether either DFOM or FAC modulates the expression of TRF1. This investigation would provide crucial insights into the potential interaction between iron and TGEV infection.
- Within the discussion section, a more comprehensive elucidation of the mechanism by which TGEV induces damage to RBCs should be undertaken. Such an elaboration would enhance the understanding of TGEV pathogenesis.
- The authors have demonstrated that TGEV infection results in injury to RBCs, subsequently leading to hypoxia in piglets. This critical point should emphasizewithin the manuscript.

Round 2
Reviewer 1 Report
Comments and Suggestions for Authors
In the revised manuscript, two major problems still have not been solved. First of all, the direct binding between virions and RBCs are not been observed. The interaction exists between other viruses and RBCs is not the reason for the direct connection between TGEV and RBCs. Secondly, the cut-off value of TGEV is 103.8 per ml, which shows the detection is unreliable. There exist various qRCP methods with much higher sensitivity of TGEV, which can be used.
Round 3
Reviewer 1 Report
Comments and Suggestions for Authors
In the manuscript, the authors have not provided the direct evidence of TGEV binding to RBCs. Moreover, the qPCR detection is of high cut-off value, which is considered as reliable. Therefore, the study is not qualified for publication.
Author Response
Comments 1: In the manuscript, the authors have not provided the direct evidence of TGEV binding to RBCs. Moreover, the qPCR detection is of high cut-off value, which is considered as reliable. Therefore, the study is not qualified for publication.
Response 1: For no available antibodies are currently available in our laboratory, TGEV binding to RBCs after in future experiments once appropriate antibodies are generated. The limit of detection for TGEV in tissue is sensitive.